# See or Guess: Counterfactually Regularized Image Captioning

## ABSTRACT

Image captioning, which generates natural language descriptions of the visual information in an image, is a crucial task in vision-language research. Previous models have typically addressed this task by aligning the generative capabilities of machines with human intelligence through statistical fitting of existing datasets. While these models demonstrate proficiency in describing the content of normal images, they may struggle to accurately describe those where certain parts of the image are obscured or edited. Conversely, humans effortlessly excel at it in this case. The weaknesses these models exhibit, including hallucinations and limited interpretability, often result in performance declines when applied to scenarios involving shifted association patterns. In this paper, we present a generic image captioning framework that leverages causal inference to make existing models more capable of interventional tasks, and counterfactually explainable. Specifically, our approach consists of two variants that utilize either total effect or natural direct effect. We incorporate these concepts into the training process, enabling the models to handle counterfactual scenarios and thereby become more generalizable. Extensive experiments on various datasets have demonstrated that our method can effectively reduce hallucinations and increase the model's faithfulness to the images, with a high portability for both small-scale and large-scale image-to-text models.

## KEYWORDS

Image Captioning, Counterfactual Causal Inference, Object Hallucination, Image-to-text Generation

## 1 INTRODUCTION

As a fundamental task in vision-language understanding research, image captioning requires models to mimic the human ability to compress huge amounts of visual information into descriptive language [3, 19, 36]. A large amount of image-to-text methods [31] have been developed, among which recent large multimodal models [12, 18, 43] perform surprisingly well in describing an image in details. Despite their good performance in real scenarios, their capabilities still differ from those of humans in interventional scenarios. For example, in Figure 1, the BLIP model [13] can generate a sentence that accurately describes the factual image at left. However, when the bicycle is masked or changed to a tree as shown in the counterfactual images, it generates incorrect descriptions such as "A man sitting on a bench in front of a river." Such errors reveal

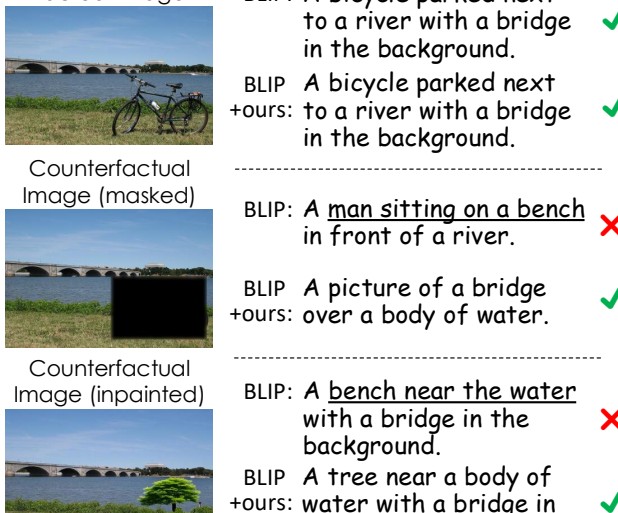

**Figure 1: An example of generated captions of different methods in the factual and two counterfactual scenarios (masked and inpainted).**

that the model might not have precisely understood the image. Instead, it may make guesses based on the common association patterns in the datasets. For example, the frequent co-occurrence of a river and a man in the dataset may lead the model to form shortcut connections and wrongly generate "man" for most images with a river.

The above analysis suggests that while current models may exhibit impressive performance, it does not necessarily imply their ability to accurately comprehend the contents of an image and generate appropriate descriptions, a capability inherently possessed by humans. Such weaknesses may result in hallucinations and hinder the interpretability of models since people cannot exactly tell which parts of the image correspond to the generated words in the text. Furthermore, when these learned models are applied to other scenarios with shifted association patterns, their performance may suffer a substantial deterioration.

To overcome the above shortcomings, we design a novel framework that integrates causal inference into any image captioning model to mitigate the shortcut correlations. Specifically, we leverage counterfactual concepts to enhance the correspondence between visual and textual characteristics. We hope that when certain regions of an image are removed, the generated text should not include the descriptions of those regions. While this idea is intuitive, it is challenging to implement for the following reasons. First, existing counterfactual models primarily focus on classification tasks [1, 8, 39]. However, we handle a generation task that necessitates the consideration of sequential impacts between words. Moreover, in our

multi-modal scenario, the influence of the image on the word can be attributed to two paths: (1) a direct influence from the image to the word, and (2) an indirect influence, where the image first impacts preceding words, which subsequently influence the current word. It presents a nontrivial challenge to distinguish between these two paths and enhance the first one to minimize hallucination while preserving linguistic fidelity.

To address the above challenges, in this paper, we first formalize the image-to-caption generation task as a sequential causal graph, where each word in the generated text is determined by both its previous words and the image. Following this causal graph, we leverage the causal concepts of total effect (TE) and natural direct effect (NDE) to discriminate different reasons for the word generation process. Then we can intervene in the cause, and enhance the correspondences between the image and words while controlling the other influential factors. Finally, we propose a counterfactually regularized image captioning framework. The main contributions of our paper are as follows:

• We propose a generic framework to counterfactually regularize image captioning models and thus make them more human-like, explainable, and robust.

• We propose two causal methods based on total effect and natural direct effect to enhance the correspondence between the visual and textual characteristics.

• We extensively experimented on various models and datasets to demonstrate the high generality and interpretability of our methods, which can effectively reduce object hallucinations and enhance the faithfulness of the model to the images.

## 2 RELATED WORK

### 2.1 Image Captioning

Image captioning, a crucial aspect of image-to-text generation [31], has evolved from convolution neural network (CNN)-based encoders and recurrent neural network (RNN)-based decoders [34, 37] to Transformer architectures [5, 11], and further into the era of vision language pretraining (VLP) models [13, 14, 42]. Recent advancements in Large Vision Language Models [12, 18, 43] have sparked renewed interest in the field. In addition, some methodologies explore the integration of multimodal representation models like CLIP [27] to furnish visual support for language models [19]. Our proposed approach has a model-agnostic nature and flexibility. Considering the notable performance of VLP models, we opt to validate the effectiveness across various architectures (decoder-only and encoder-decoder) and model scales by employing ClipCap [19], BLIP [13], and BLIP2 [12] as backbone models.

### 2.2 Object Hallucination in Image Captioning

Alleviating hallucination of image captioning models does not solely hinge on improved image perception capability but also on factors like over-reliance on language priors or biases during sequence generation [25, 28], potentially leading to guesswork that is not faithful to the image. Researchers [28] thus propose utilizing the CHAIR metric to quantify hallucination occurrence. Some efforts have been made to reduce model reliance on common or biased co-occurrences by adjusting object label co-occurrence statistics [4].

Other methods maintain semantic consistency to reduce object hallucinations by learning consensus representations through aligning scene and language graphs [40], or by aligning textual tokens and visual objects using masked language modeling [6]. However, these methods may blur semantic and visual alignments and overly rely on dataset co-occurrence patterns, harming interpretability and performance in real scenarios. Our approach considers causality, aiming to establish the correct vision-to-language guiding relationship during the generation process.

### 2.3 Counterfactual Causal Inference

Causal inference seeks to unravel the causal relationships and underlying mechanisms driving observed outcomes [2, 7, 23]. Moreover, counterfactual causal inference offers a framework to enhance [1, 32] and explain [8, 9] models in counterfactual scenarios. However, the majority of these counterfactual-related works are tailored for classification tasks, such as image classification [1, 8, 39], representations learning [32, 41], or visual question answering [10, 15, 20], rather than for generation tasks. Classification tasks exhibit a deterministic correspondence between input and output, whereas, in the generation process, the counterfactual image and preceding generated tokens collectively influence the subsequent token generation, creating an effect propagation. Some researchers [35] have performed Maximum Likelihood Estimation (MLE) on interventional distribution to reduce spurious correlations learned by models due to observed confounding factors. However, the applicability of their framework is limited to the strong ignorability assumption and lacks causal analysis in multi-modal scenarios. Capturing this causal correspondence [22] is challenging, especially in multimodal scenarios, and has thus received little attention in prior literature. In this paper, we endeavor to leverage counterfactual causal inference to tackle this challenge and gain deeper insights into a model's generation behavior.

## 3 PRELIMINARIES: A CAUSAL LOOK AT IMAGE CAPTIONING

This section presents the fundamental concepts and notations of causal inference [7, 22] and how we apply it in image captioning. In the following, capital letters, *i.e.*, cause $X$, Mediator $M$ and Effect $Y$, represent random variables. The values or subscripts of these random variables indicate their observed values.

As for image captioning, a model is used to process an input image $I$ and produce a corresponding textual description, *i.e.*, a sequence $S = (s_1, s_2, \ldots, s_L)$, where $s_i$ is a token in the sequence and $L$ is the sequence length. The sequence of the preceding tokens of $s_j$ is denoted as $S_{<j} = (s_1, s_2, \ldots, s_{j-1})$. We will later present how to treat these variables from a causal perspective.

### 3.1 Causal Graph

A causal graph describes the causal relations between different random variables in a graph manner [24]. In a causal graph $\mathcal{G} = \{\mathcal{V}, \mathcal{E}\}$, a node $v \in \mathcal{V}$ represents a variable and a directed edge $e \in \mathcal{E}$ represents a causal relationship between variables. The *direct effect* means that there is an edge between two variables, *e.g.*, in Figure 2(a), $X$ has a *direct effect* on $Y$. The *indirect effect* means that

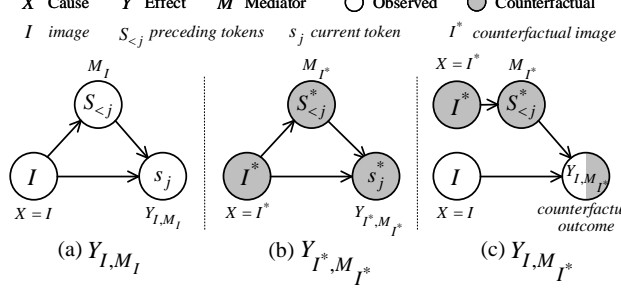

**Figure 2: Illustration of causal graphs and counterfactual causal effect notations.**

two variables are not directly linked, but are connected via some *mediator* variables, *e.g.*, $X$ has a *indirect effect* on $Y$ if $X \rightarrow M \rightarrow Y$.

Considering the process of auto-regressive generation, at each step, the current token $s_j$ is determined by all the preceding tokens $S_{<j}$, and the visual information of the input image $I$ as well. As shown in Figure 2(a), at step $j$, $S_{<j}$ is influenced by $I$, and $s_j$ is jointly determined by $I$ and $S_{<j}$. We use $Y_{I,M_I} = Y(X = I, M = M_I)$ to denote the probability of token $s_j$ when the cause $X$ is set to $I$ and the mediator $M$ is set to $M_I = S_{<j}$.

### 3.2 Counterfactual Causal Effects

In causal inference, counterfactual causal effects compare hypothetical outcomes under factual and counterfactual treatments [2, 23].

As shown in Figure 2(b), the value of the counterfactual of variable $X$ is equal to the counterfactual image $I^*$, where $I^*$ is created by intervening in the factual image $I$. The hypothetical outcome of $Y$ is denoted as $Y_{I^*,M_{I^*}} = Y(X = I^*, M = M_{I^*})$, where the mediator $M_{I^*} = S_{<j}^*$. The total effect (TE) is the difference between two hypothetical conditions: one being factual transition where $X = I$ (under treatment, corresponding to Figure 2(a)) and the counterfactual being $X = I^*$ (under no-treatment, corresponding to Figure 2(b)). Mathematically, the total effect can be expressed as

$$\text{TE}_{I,I^*} = Y_{I,M_I} - Y_{I^*,M_{I^*}}. \tag{1}$$

$\text{TE}_{I,I^*}$ measures the effect of all factors (*i.e.*, direct and indirect effects) resulting from changing image $I$ to $I^*$.

Further, intervening both $X$ and $M$ allows the total effect to be decomposed into two components, namely the natural direct effect (NDE) and the total indirect effect (TIE). Unlike TIE that focuses on the effect brought by changes in the mediator $M$, NDE is the effect of $X$ on $Y$ that results solely from changes in $X$, without any influence from $M$, which can be denoted as

$$\text{NDE}_{I,I^*} = Y_{I,M_{I^*}} - Y_{I^*,M_{I^*}}. \tag{2}$$

The first term $Y_{I,M_{I^*}}$ corresponds to Figure 2(c), which keeps $X = I$ and conducts intervening on $M$ via $I^*$ to form a counterfactual outcome $Y_{I,M_{I^*}}$. The second term $Y_{I^*,M_{I^*}}$ corresponds to Figure 2(b). Formula 2 describes the variation of $Y$ when $X$ is changed from $I$ to its counterfactual $I^*$ while $M$ is held constant at $M(X = I^*)$.

This paper explores how to use TE or NDE to reduce object hallucination in image captioning and improve interoperability.

## 4 COUNTERFACTUAL REGULARIZATION

In this section, we introduce how to construct counterfactual data, propose our framework, and design two counterfactual regularization losses that can be universally applied to existing image captioning models.

### 4.1 Constructing Counterfactual Data

Collecting or generating counterfactual images for the factual ones is difficult. However, by adding a mask, it is easy to achieve minimal changes to the original image when constructing the counterfactual one, which can be regarded as an approximation of the idealized counterfactual image. Specifically, we construct counterfactual images by leveraging datasets that have labeled bounding boxes for corresponding phrases in the image captions. As shown in Figure 3(a), we first select the entity to intervene ($\tilde{S}$), *e.g.*, "black poodle". Then we identify its corresponding region to intervene ($r$) in the image based on the labeled bounding boxes in the datasets. A black mask is used to replace the region $r$ to create a counterfactual image $I^*$. If the entity to intervene corresponds to more than one bounding box (*e.g.*, when $\tilde{S}$ is "a group of people"), all related regions will be masked. Next, we employ the initial image captioning model to generate a counterfactual caption $S^*$. The counterfactual captions are used to model $S_{<j}^*$ and $s_j^*$ in the causal graphs (Figure 2), *i.e.*, modeling the generated words for the counterfactual image $I^*$. Please note that the goal of $S^*$ is to facilitate the estimation of causal effects, instead of being used as a ground-truth caption for counterfactual images. Thus, it is much easier to obtain compared with ground-truth labels for counterfactual images. Accordingly, we have $(I, S, \tilde{S}, I^*, S^*)$ prepared for dataset $\mathcal{D}$.

### 4.2 Our Framework

We propose a framework by incorporating negative log-likelihood (NLL) loss $\mathcal{L}_{\text{NLL}}$ with TE or NDE regularization loss, *i.e.*, $\mathcal{L}_{\text{TE}}$ or $\mathcal{L}_{\text{NDE}}$, which will be described later. Formally, the vanilla negative log-likelihood (NLL) loss is as follows:

$$\mathcal{L}_{\text{NLL}} = - \sum_{(I,S) \in \mathcal{D}} \sum_{i=1}^{L} \log f_\theta(s_i \mid I, S_{<i}), \tag{3}$$

where $f_\theta(\cdot)$ refers to the model that takes the image and preceding text sequence $S_{<i}$ as input and outputs a probability distribution on the vocabulary to generate the next token $s_i$, with parameters $\theta$.

We add the counterfactual regularization loss to allow the model to learn together with the NLL loss. A hyperparameter $\alpha$ determines the weight of losses, ensuring balanced optimization. The final loss is denoted as:

$$\begin{aligned} \mathcal{L}_1 &= \alpha \mathcal{L}_{\text{NLL}} + (1 - \alpha) \mathcal{L}_{\text{TE}}, \\ \mathcal{L}_2 &= \alpha \mathcal{L}_{\text{NLL}} + (1 - \alpha) \mathcal{L}_{\text{NDE}}. \end{aligned} \tag{4}$$

The whole optimization includes two stages: (1) training the model with vanilla NLL loss (Formula 3); (2) training the model with either $\mathcal{L}_1$ or $\mathcal{L}_2$ (Formula 4) using the constructed counterfactual images and their corresponding generated counterfactual captions.

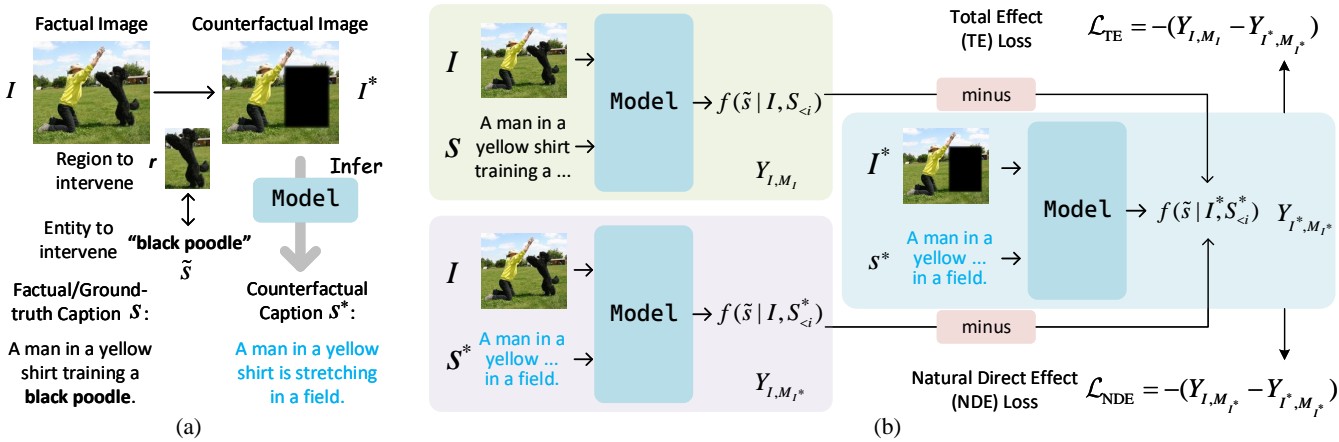

**Figure 3: Our framework of counterfactual regularization. (a) shows how to prepare counterfactual images and captions by an example. (b) illustrates how the TE loss and NDE loss are calculated in the example. Counterfactual captions are in blue. The phrase corresponding to the image region in the mask is "black poodle". Best viewed in color.**

### 4.3 Total Effect Regularization

When the region corresponding to "black poodle" is masked in the image (Figure 3), we hope the model will significantly lower the generation probabilities of the words "black poodle" to reduce hallucination. To achieve this goal from a causal perspective, we maximize the total effect of changing $I$ to $I^*$ on the generation of "black poodle" (*i.e.*, $\tilde{S}$), which is given in Formula 1. Maximizing this total effect can be fulfilled by minimizing the following total effect (TE) loss:

$$
\mathcal{L}_{\text{TE}} = -\sum_{(I,S)\in\mathcal{D}} \sum_{j=1}^{L_{\tilde{S}}} \left[ \log f_\theta(\tilde{s}_j \mid I, S_{<p+j}) \right.
$$
$$
\left. -\frac{1}{L_{S^*}} \sum_{i=1}^{L_{S^*}} \log f_\theta(\tilde{s}_j \mid I^*, S^*_{<i}) \right], \tag{5}
$$

where the first part corresponds to $Y_{I,M_I}$ in Formula 1 and calculates the likelihood of generating $\tilde{S}$ (*e.g.*, "black poodle") given the factual image $I$ and previously generated words, and the second part estimates $Y_{I^*,M_{I^*}}$ in Formula 1 with the likelihood of generating $\tilde{S}$ at any position given the counterfactual image $I^*$ and preceding tokens $S^*_{<i}$ that are generated from $I^*$. Here, $\tilde{s}_j$ denotes the $j$-th token of the entity to intervene $\tilde{S}$, of which the length is $L_{\tilde{S}}$. $p$ represents the index of the first position where the entity to intervene $\tilde{S}$ appeared in the ground truth or factual caption (*e.g.*, if "black" and "poodle" are the 9th and 10th words, then $p = 9$). $L_{S^*}$ is the length of counterfactual caption $S^*$. The specific occurrence position of the entity to intervene $\tilde{S}$ is explicit in the ground truth, while it may not necessarily appear in the counterfactual caption. Therefore, we need to estimate the probability of their occurrence using the average value.

In the example shown in Figure 3, the word "black poodle" is the entity to intervene ($\tilde{S}$). We estimate the first term by the probability of generating each token in "black poodle" at the position it appeared in the factual caption, *i.e.* using the preceding tokens "A man in a yellow shirt training a". The second term is calculated by the average probabilities of generating each token in "black poodle" at any position in the counterfactual caption "A man in a yellow

shirt is stretching in a field.", where the preceding tokens are those before each step.

### 4.4 Natural Direct Effect Regularization

To improve the visual perception ability of the model, another option is to maximize the natural direct effect (NDE) rather than the total effect (TE). The natural direct effect is to measure the direct effect resulting from changes in the image. As shown in Formula 2, the first part is to calculate the likelihood of generating each token in the entity to intervene $\tilde{S}$ at any position, from the image $I$ and the preceding tokens $S^*_{<i}$ that have been generated from $I^*$. Whereas, the second part is the likelihood of generating $\tilde{S}$ at any position from the counterfactual image $I^*$ and preceding tokens $S^*_{<i}$ that are generated from $I^*$, which is the same as the second part of TE. Formally, we calculate NDE loss as:

$$
\mathcal{L}_{\text{NDE}} = -\sum_{(I,S)\in\mathcal{D}} \sum_{j=1}^{L_{\tilde{S}}} \left[ \frac{1}{L_{S^*}} \sum_{i=1}^{L_{S^*}} \left( \log f_\theta(\tilde{s}_j \mid I, S^*_{<i}) \right. \right.
$$
$$
\left. \left. -\log f_\theta(\tilde{s}_j \mid I^*, S^*_{<i}) \right) \right]. \tag{6}
$$

The first part here looks simpler than that in TE loss because both the first part and the second part in NDE loss average the probabilities over any position in $S^*$ whose length is $L_{S^*}$.

Figure 3(b) presents an example. The first term is estimated by the probability of generating each token in "black poodle" at any position in the counterfactual caption "A man in a yellow shirt is stretching in a field.", but with the factual image $I$ as input. The second term is again the average probabilities of generating each token in "black poodle" at any position in the counterfactual caption with the counterfactual image $I^*$ as input. By maximizing the NDE effect, the direct influence of the image is enhanced, thereby the model is more inclined to see the image and generate the correct next token, rather than to guess it.

**Table 1: Evaluation results on counterfactual images with masks. CH.$_s$ (CHAIR$_s$), P$_{@5}$ (Precision$_{@5}$), and nDCG$_{@5}$ are automatic measures for evaluating hallucination. Faith. (Faithfulness) and Overall denote results given by human judges. The best result is highlighted in bold, while the second best is underlined.**

| Methods | Flickr30k Entities | | | | | MSCOCO | | | | |
|---|---|---|---|---|---|---|---|---|---|---|
| | CH.$_s$ ↓ | P$_{@5}$ | nDCG$_{@5}$ | Faith. | Overall | CH.$_s$ ↓ | P$_{@5}$ | nDCG$_{@5}$ | Faith. | Overall |
| **ClipCap** | 20.45 | 80.08 | 79.97 | 0.320 | 0.447 | 64.05 | 36.02 | 36.01 | 0.687 | 0.713 |
| +ObjL [4] | 21.18 | 79.16 | 79.07 | 0.353 | 0.453 | 64.64 | 35.62 | 35.58 | 0.653 | 0.727 |
| +ObjMLM [6] | 25.37 | 75.07 | 74.97 | 0.140 | 0.313 | 70.07 | 29.89 | 29.91 | 0.533 | 0.547 |
| +TE (ours) | 19.78 | 80.48 | 80.39 | 0.373 | 0.467 | 63.58 | 36.32 | 36.35 | 0.753 | 0.807 |
| +NDE (ours) | **19.64** | **80.53** | **80.51** | **0.400** | **0.493** | **63.04** | **36.55** | **36.65** | **0.760** | **0.820** |
| **BLIP** | 12.14 | 88.00 | 88.00 | 0.740 | 0.793 | 33.70 | 66.17 | 66.22 | 1.167 | 1.200 |
| +ObjL [4] | 10.61 | 89.17 | 89.19 | 0.613 | 0.767 | 33.07 | 67.12 | 67.08 | 1.187 | 1.107 |
| +ObjMLM [6] | 10.11 | 89.31 | 89.45 | 0.687 | 0.787 | 33.90 | 65.67 | 65.77 | 1.113 | 1.180 |
| +TE (ours) | 10.23 | 89.63 | 89.68 | 0.767 | 0.827 | 31.10 | 68.49 | 68.58 | 1.213 | 1.267 |
| +NDE (ours) | **9.53** | **89.83** | **89.93** | **0.873** | **0.913** | **30.43** | **69.24** | **69.33** | **1.247** | **1.273** |
| **BLIP2** | 8.01 | 91.95 | 91.96 | 0.807 | 0.847 | 30.28 | 69.91 | 69.88 | 1.227 | 1.233 |
| +ObjL [4] | 8.02 | 91.90 | 91.96 | 0.847 | 0.887 | 30.26 | 70.19 | 70.13 | 1.133 | 0.947 |
| +ObjMLM [6] | 8.12 | 92.00 | 92.01 | 0.800 | 0.867 | 34.84 | 65.23 | 65.19 | 1.140 | 1.100 |
| +TE (ours) | 7.61 | 92.09 | 92.14 | **0.867** | **0.913** | 29.60 | 70.54 | 70.49 | 1.340 | 1.273 |
| +NDE (ours) | **7.51** | **92.21** | **92.24** | 0.860 | 0.880 | **29.26** | **70.70** | **70.68** | **1.353** | **1.280** |

## 5 EXPERIMENTS

In this section, we conduct extensive experiments to evaluate the capability of our model for alleviating object hallucination, reducing biases in training data, and interpreting the correspondence between captions and image regions.

### 5.1 Experiment Setup

*5.1.1 Datasets.* To evaluate the effectiveness of our model, we construct counterfactual images and captions as mentioned in Section 4.1. We choose Flickr30k Entities [26] and MSCOCO [17] as our datasets, which have high-quality image annotations for constructing masked counterfactual images.

**Flickr30k Entities** (Flickr) is built upon the existing Flickr30k dataset [38] that contains 31,783 images. The dataset provides 244k coreference chains and 276k manually annotated bounding boxes within the images. We use the original split of this dataset. Entities that occur more than once within a caption are removed to avoid confusion. After pre-processing, the final dataset consists of 29k/1k/1k samples for training/validation/test, respectively.

**MSCOCO** (COCO) consists of more than 328k images with annotated objects, phrases, and relationships. We adopt the Karpathy split of the MSCOCO dataset and coreference relationships in the annotations are utilized to establish correspondences between the image regions and phrases (entities) to intervene in the captions.

Both datasets are composed of diverse phrase categories, where the Flickr dataset covers over 1,000 categories, while the COCO dataset is more concentrated on 80 categories.

*5.1.2 Backbones and Baselines.* Our proposed counterfactual regularization losses are model-agnostic and can be applied to various models. We conduct experiments on three backbones: Clip-Cap [19], BLIP [13] and BLIP2 [12], which respectively serve as

representative models for decoder-only, encoder-decoder, and multimodal large language model architectures. In addition to the above three image captioning baselines, we compare our methods with another two methods that aim to alleviate the object hallucination in image captioning: (1) **ObjL** [4] utilizes object labels as training augmentation to diminish models' object bias on hallucination; (2) **ObjMLM** [6] conducts a whole object mask to mitigate object hallucination in masked language modeling. More implementation details are in supplementary pages.

*5.1.3 Evaluation Methodology.* Compared to baselines, our methods are expected to significantly reduce object hallucination on counterfactual test sets while maintaining the generation ability on factual test sets (it is not trivial due to different distributions between training and test sets). We employ both automatic and human evaluation in our experiments for convincing conclusions. **Automatic Evaluation:** We evaluate hallucination by using:

- **CHAIR$_s$** [28]: It measures whether models generate a masked phrase, *i.e.*, phrases whose corresponding regions have been masked in the counterfactual image:

$$\text{CHAIR}_s = \frac{|\{\text{captions with hallucinated objects}\}|}{|\{\text{all captions}\}|}, \quad (7)$$

where a lower CHAIR$_s$ score indicates a reduced presence of hallucination or increased faithfulness.

- Ranking-based Metrics: We generate a set of five candidate captions with the highest probability of being generated for a given counterfactual image, among which the ones without the masked phrase are regarded as positive while those with the masked phrase are as negative. **Precision$_{@5}$** and **nDCG$_{@5}$** are employed[1] to assess the object hallucination in fine-grained.

---

[1]https://github.com/microsoft/rankerEval

Table 2: Evaluation results on factual images without masks. BLEU-4, ROUGE-L, CIDEr are automatic measures for evaluating generation quality. Faith. (Faithfulness) and Overall denote content accuracy and overall caption quality given by human judges. The best result is highlighted in bold, while the second best is underlined.

| Methods | Flickr30k Entities | | | | | MSCOCO | | | | |
|---|---|---|---|---|---|---|---|---|---|---|
| | BLEU-4 | ROUGE-L | CIDEr | Faith. | Overall | BLEU-4 | ROUGE-L | CIDEr | Faith. | Overall |
| **ClipCap** | 23.38 | 48.33 | 57.09 | 0.947 | 0.793 | 28.79 | 52.75 | 126.92 | 1.360 | 1.253 |
| +ObjL [4] | 23.58 | 48.17 | 58.13 | 0.927 | 0.787 | 26.67 | 49.40 | 115.15 | 1.373 | 1.173 |
| +ObjMLM [6] | 17.01 | 43.64 | 32.42 | 0.793 | 0.813 | 19.25 | 46.76 | 73.82 | 1.040 | 1.013 |
| +TE (ours) | **24.03** | **48.92** | **59.08** | **0.973** | **0.860** | **28.94** | **52.98** | **127.27** | 1.413 | **1.300** |
| +NDE (ours) | 23.32 | 48.05 | 58.69 | 0.960 | 0.840 | 28.77 | 52.89 | 125.70 | **1.433** | 1.280 |
| **BLIP** | 37.14 | 56.67 | 95.40 | 1.433 | 1.260 | 34.63 | **56.83** | 153.39 | 1.873 | 1.593 |
| +ObjL [4] | 36.93 | 56.37 | 92.72 | 1.420 | 1.147 | 33.00 | 56.66 | 148.00 | 1.840 | 1.300 |
| +ObjMLM [6] | 35.61 | 56.56 | 94.88 | 1.420 | 1.253 | 31.71 | 65.55 | 133.67 | 1.713 | 1.540 |
| +TE (ours) | **37.28** | **56.77** | 95.40 | 1.447 | 1.300 | 34.65 | 56.82 | 153.37 | 1.900 | 1.620 |
| +NDE (ours) | 37.00 | 56.68 | **95.45** | **1.473** | **1.307** | **34.66** | **56.83** | **153.73** | **1.920** | **1.640** |
| **BLIP2** | 37.61 | 58.11 | 103.41 | 1.473 | 1.373 | 34.72 | 58.13 | 154.17 | 1.880 | 1.553 |
| +ObjL [4] | 34.30 | 56.61 | 93.43 | 1.473 | 1.240 | 29.81 | 55.13 | 135.42 | 1.820 | 1.340 |
| +ObjMLM [6] | 36.04 | 56.94 | 94.76 | 1.480 | 1.220 | 34.06 | 57.39 | 151.51 | 1.800 | 1.453 |
| +TE (ours) | **37.64** | **58.24** | **103.68** | 1.520 | 1.393 | 34.80 | 58.24 | 154.77 | 1.933 | **1.647** |
| +NDE (ours) | 37.56 | 58.11 | 102.63 | **1.533** | **1.427** | **34.88** | **58.34** | **155.14** | **1.947** | 1.613 |

We adopt **BLEU** [21], **ROUGE-L** [16], and **CIDEr** [33] to measure the quality of generated captions on factual image test sets. We do not evaluate the generation quality of counterfactual images using automatic evaluation due to the lack of ground-truth captions. To compensate for this, the quality of captions generated for counterfactual images is analyzed by using human evaluation.

**Human Evaluation:** To verify whether the automatic measurements are consistent with human experiences, we further conduct a user study. First, we randomly sample 50 factual images from the Flickr and 50 from the COCO dataset. We then create counterfactual images for the 100 factual images and collect top-generated captions from all methods for both factual and counterfactual images. We conduct human evaluations on the 100 factual images and 100 counterfactual images. For each image, we shuffle the generated captions and make the methods anonymous when presented with an image to ensure a fair comparison. Three human assessors majored in English with the age range from 23 to 25, are hired to rate the captions on a 3-level Likert scale from 0 to 2 in two aspects:

• **Faithfulness** measures the degree to which a caption accurately represents the content of the image;

• **Overall** means the overall quality of a caption.

Finally, we calculate the Fleiss' Kappa among their assessments which results in 0.43, meaning a moderate level of agreement. We use their average values as the results.

## 5.2 Evaluation Results on Counterfactual Images

We first compare our proposed models with all baselines on counterfactual test sets in terms of both hallucination and overall generation quality. The results are shown in Table 1, where all methods with the same backbone are grouped for clarity.

**Automatic evaluation**. In terms of the automatic metrics of measuring object hallucination in Table 1 (CH.$_s$, P$_{@5}$, nDCG$_{@5}$), our proposed counterfactually regularized methods consistently exhibit superior performance over all baselines on both datasets, demonstrating their effectiveness in mitigating object hallucination. Our NDE regularization performs better than the TE ones. This indicates that maximizing the direct effect of image content on the generated tokens helps build a more precise alignment between visual regions and their corresponding entity phrases. Baselines ObjL and ObjMLM do not always alleviate hallucination effectively, e.g. , they exhibit more hallucinations on the ClipCap backbone on the two datasets. In comparison, our methods that regularize the causal effect consistently reduce hallucination in terms of different backbones, datasets, and measures. This demonstrates the effectiveness of adopting a causal perspective when handling hallucinations. Further experiments confirm this will not be affected by different decoding strategies (please see Section A in supplementary pages).

**Human evaluation**. Human evaluation results in Table 1 show that our methods perform the best regarding both reducing hallucination (Faith.) and overall generation quality (Overall). Moreover, NDE wins more times than TE, and these two methods that we propose are consistently better than baselines ObjL and ObjMLM. Overall, the human evaluation results are consistent with the automatic evaluation results in terms of hallucination and additionally reveal the good generation quality of our methods.

## 5.3 Evaluation Results on Factual Images

We compare all methods on factual test images to investigate 1) whether our regularization methods compromise any generation capability and 2) whether our method can reduce hallucination on factual images. As shown in Table 2, our proposed methods

Factual image     Counterfactual image          Factual image     Counterfactual image          Factual image   Counterfactual image

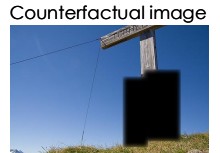
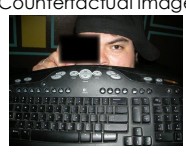
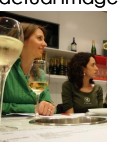
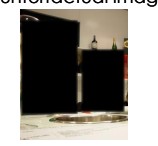

GT: Two people stand next to a wood cross on a grassy hill.

BLIP: A person is pulling a rope from a wooden sign.

BLIP+TE: A wooden sign on a grassy hill with a blue sky in the background.

BLIP+NDE: A wooden cross on a grassy hill with a blue sky in the background.

GT: A man in a black cap is holding a computer mouse up to one of his eyes as he holds a computer keyboard in front of his face.

ClipCap: A man is typing on a computer keyboard.

ClipCap+TE: A man is sitting in front of a computer keyboard.

ClipCap+NDE: A man with a black hat and a black keyboard.

GT: Two women sitting at a table looking at another person with a shocked look.

BLIP: A cat sitting on top of a counter next to a bottle of wine.

BLIP+TE: A couple of black screens sitting on top of a counter.

BLIP+NDE: A couple of black screens sitting on top of a counter.

**(a) Cases on Flickr and MSCOCO dataset**

Factual image     Counterfactual image          Factual image     Counterfactual image          Factual image   Counterfactual image

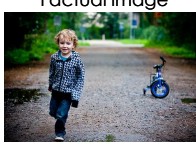
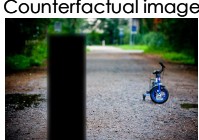
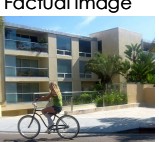
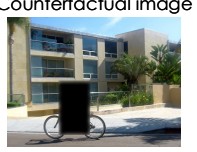
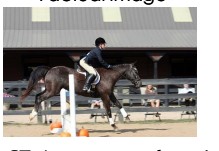
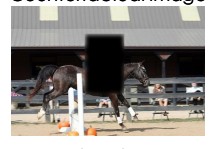

GT: A little girl walking away from her bicycle and walking down the street.

BLIP: A little boy is walking down a path.

BLIP+TE: A child's bike is parked on the side of the road.

BLIP+NDE: A child's bike is parked on a gravel path.

GT: A young girl rides her bike by an apartment building.

BLIP: A man is standing next to a bicycle.

BLIP+TE: A bicycle leaning against a wall.

BLIP+NDE: A bicycle is parked on the side of the road.

GT: A woman on a horse jumps an obstacle.

BLIP: A man in a suit is riding a horse.

BLIP+TE: A horse is jumping over an obstacle.

BLIP+NDE: A horse is jumping over an obstacle.

**(b) Cases on gender biased dataset**

Factual image   Counterfactual image          Factual image   Counterfactual image          Factual image   Counterfactual image

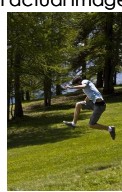
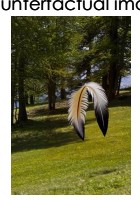
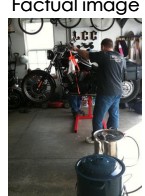
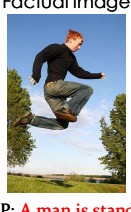
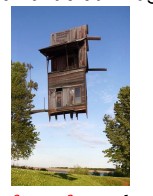

BLIP2: A person flying a kite in a field.

BLIP2+TE: A feather flying in a field.

BLIP2+NDE: A feather flying in a field.

BLIP: Two men working on a motorcycle in a garage.

BLIP+TE: Two men working in a garage.

BLIP+NDE: Two men working in a garage.

BLIP: A man is standing in front of a tree house.

BLIP+TE: A tree house is suspended in the air.

BLIP+NDE: A tree house in the middle of a field.

**(c) Cases on inpainting dataset**

**Figure 4: Examples of generated captions by different methods on some masked or inpainted counterfactual images. Phrases highlighted in red are hallucinations that do not exist in the counterfactual image.**

**Table 3: Error rate (out of 2,034 samples) of predicting female as male on two test set. We show the number of samples with errors in parentheses.**

| Error Rate | Factual Image | Counterfactual Image |
|---|---|---|
| BLIP | 13.91% (283) | 38.25% (778) |
| BLIP+TE | 13.96% (284) | **34.12% (694)** |
| BLIP+NDE | **13.27% (270)** | 34.27% (697) |

achieve comparable or superior performance to all the baselines on both datasets in terms of automatic metrics for evaluating generation quality (BLEU, ROUGE-L, CIDEr). Human evaluation results also show that our methods can consistently outperform all baselines in reducing hallucination (Faithfulness) and increasing overall generation quality (Overall). This indicates that our methods can

significantly reduce hallucinations on counterfactual images without scarification in generation performance on factual images.

## 5.4 Evaluation over Biased Datasets

It would be interesting to investigate whether the proposed methods perform better when the test data has a biased distribution of some entities from training data. Therefore, we construct a biased dataset from Flickr30k Entities. First, we do statistics of all the captions and find that 9,893 captions contain male-related words, such as "man/men" and "boy/boys", and 5,963 captions contain female-related words. We then reconstruct a training set consisting of 8,942 male, 1,962 female, and 14,838 other captions, where the ratio of male to female is about 5:1. We reverse the ratio to reconstruct the test set, which consists of 481 males, 2,034 females, and 500 other captions. The validation set is constructed with a similar size and

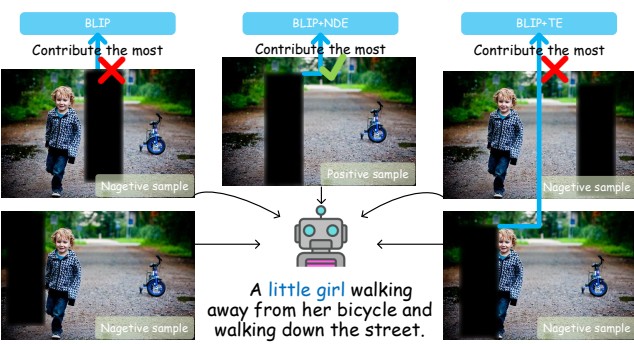

**Figure 5: Illustration of the experiment design to evaluate interpretability.**

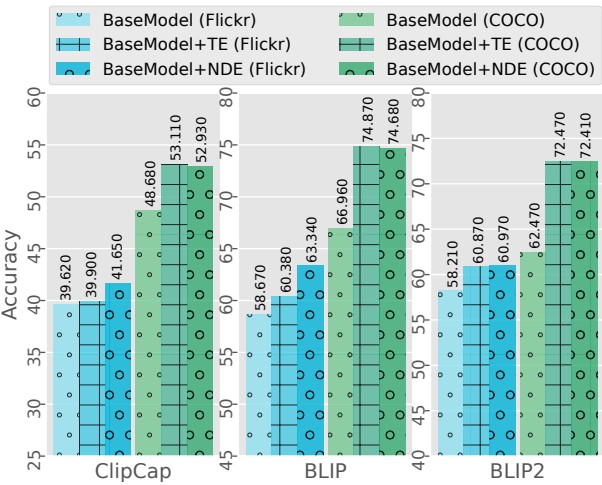

**Figure 6: The interpretability performance of different models by identifying the correct masked counterfactual image.**

recipe as the test set. Finally, we train a BLIP model and evaluate its performance on the biased test set.

When examining entities related to gender only, the error rates are presented in Table 3. The results indicate that BLIP+NDE model outperforms BLIP in terms of lower error rates for both factual and counterfactual images. The results imply that models with our proposed methods are more robust in handling biased datasets.

## 5.5 Quantitative Analysis

We present some examples of the generated captions on counterfactual images in Figure 4(a), and more in supplementary pages. Overall, our methods perform better in understanding counterfactual images, avoiding generating captions containing masked information. Instead, they describe what are indeed presented in the images, such as "a wooden cross" and "a couple of black screens". Conversely, the baseline model without counterfactual regularization often guesses incorrectly. We also present some examples of generated captions for counterfactual images in the biased dataset in Figure 4(b). The baseline model often incorrectly guesses a "man" or "boy" behind the mask, whereas our models describe other objects that are present in the image, such as "a child's bicycle" and "a horse".

We further utilize a Latent Diffusion Model [29] to inpaint the masked region with a counterfactual object. As shown in Figure 4(c), an intriguing observation is that the baseline model occasionally hallucinates "a person" in the inpainted image, despite the absence of any human presence in the image. This may be caused by the shortcut connections it learned from the training data, where our methods can robustly avoid this and correctly describe "a feather" or "a tree house" that can be seen in the inpainted images.

## 5.6 Evaluation of Interpretability

In this experiment, we compare different image captioning models in terms of interpretability. The basic idea is that an interpretable model should generate a noun phrase by using its corresponding region. For example, when generating the phrase "little girl", the region that contains a little girl should contribute the most to the model generation compared with other regions. The contribution of a region is measured by using an efficient and effective explanation method CXPlain [30]. More specially, for each noun phrase, we first identify its corresponding region (positive sample) and then

randomly generate four incorrect regions with the same size (negative samples). Next, we rank the five regions by using CXPlain [30], which assigns a higher contribution score to a region if removing this region results in a larger change in the model's loss. A model is considered interpretable if the positive sample has the highest contribution. Figure 5 shows an example, where the correct region (the region with a little girl) contributes the most in generating the phrase "little girl" for the Blip+NDE model, demonstrating the interpretability of the model. In comparison, for other models (Blip and Blip+TE), the correct region does not have the highest contribution to generating "little girl".

To show the average results over our test set, we use accuracy as the measure. Accuracy refers to the percentage of cases in which the positive sample has the highest contribution. Figure 6 shows that the TE method consistently performs better than the backbone model without regularization. The NDE method outperforms the TE method across backbone models on Flickr by a significant margin, while they perform comparably on MSCOCO. This suggests that our proposed counterfactual regularization is effective in enhancing interpretability and our proposed NDE method is the most effective.

## 6 CONCLUSION

This paper proposes using counterfactual causal effects to model the relationship between vision and language. We employ two counterfactual regularization methods based on the concepts of total effect (TE) and natural direct effect (NDE) to improve image captioning models. Experimental results consistently show the superiority of our methods over baselines in terms of alleviating hallucination across different backbones and datasets. The NDE method performs the best in generating faithful captions for counterfactual images and accurately interpreting the most relevant image regions corresponding to a phrase in a caption. In the future, we plan to integrate the counterfactual regularization methods into more complicated multimodal generation scenarios with both image and text as input, such as visual question answering and multimodal dialogue.

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
