# OpenReview forum: "See or Guess: Counterfactually Regularized Image Captioning"
_acmmm.org/ACMMM/2024/Conference — MM2024 Poster_

### Official Review · Reviewer_GZrh · 2024-05-11

**Rating:** 3
**Confidence:** 4

**Summary:**

This paper proposes a universal image captioning framework that leverages counterfactual causal inference to enhance the correspondence between visual and textual features, thereby improving the explainability and robustness of existing models when processing counterfactual scenarios. See strengths and limitations.

**Strengths:**

1. The paper introduces two causal methods based on total and natural direct effects, and integrates them into the training process to enhance the capability of models in processing counterfactual scenes, making them more explainable and robust.
2. Experimental results demonstrate that this method can effectively reduce illusions and increase the faithfulness of models to images, possessing high transferability and suitability for different sizes of image-to-text models.
3. The writting and the figures are clear and easy to read.

**Limitations:**

1. Some important references are missing:
[1] Le T, Lal V, Howard P. Coco-counterfactuals: Automatically constructed counterfactual examples for image-text pairs[J]. Advances in Neural Information Processing Systems, 2024, 36.
[2] Yang X, Zhang H, Cai J. Deconfounded image captioning: A causal retrospect[J]. IEEE Transactions on Pattern Analysis and Machine Intelligence, 2021, 45(11): 12996-13010.
[3] Fei Z. Efficient modeling of future context for image captioning[C]//Proceedings of the 30th ACM International Conference on Multimedia. 2022: 5026-5035.
2. It has not been tested on LLM models, which may affect the scalability of the method. Validation on larger models might be necessary.
3. The automatic assessment indicators used in the document, such as CHAIRs and P@5, may not truly reflect the performance of the model on counterfactual image, due to the dataset bias.
4. It does not consider the impact of distribution differences between test and training data. Such differences may affect the generalization ability of the model and should be taken into account during evaluation. Testing the method on more and different datasets could provide more comprehensive results, e.g. training on COCO and testing on Flickr30k.
5. It uses basic masking techniques to generate counterfactual images, which can only simulate a limited number of counterfactual scenarios. For example, replace some region with counterfactual elements.

**Suitability:**

3

---

### Official Review · Reviewer_Enuu · 2024-05-14

**Rating:** 5
**Confidence:** 3

**Summary:**

The paper addresses the challenge of accurately describing images, especially in scenarios where parts of the image are obscured or edited. Current image captioning models struggle with this issue due to problems like hallucinations and limited interpretability. The paper proposes leveraging causal inference in existing image-to-caption models. By incorporating concepts like total effect and natural direct effect into the training process, the models become more generalizable and capable of handling counterfactual scenarios. The main contributions include proposing a generic framework for counterfactually regularizing image captioning models, introducing two causal methods to enhance correspondence between visual and textual characteristics, and demonstrating effectiveness in reducing object hallucinations and improving model faithfulness to images.

**Strengths:**

The paper's strength lies in its innovative approach of incorporating causal inference into image captioning models to enhance interpretability and performance in handling counterfactual scenarios. The use of counterfactual data construction, where minimal changes are made to original images through masking, allows for improved model generalizability and robustness in generating captions for altered or obscured images. The paper's emphasis on addressing issues like object hallucinations and improving model faithfulness to images demonstrates its practical relevance and potential applications in various image captioning tasks. The paper is well-presented and organized, making it easy to read. Additionally, Figure 4 effectively illustrates how the models work.

**Limitations:**

The paper's limitation lies in the challenge of distinguishing between direct and indirect influences of the image on word generation in a multi-modal scenario. This complicates the enhancement of the direct influence to minimize hallucinations while maintaining linguistic fidelity. Additionally, I would suggest that the authors elaborate more on precision and nDCG in the "Experiments" section.

**Suitability:**

3

---

### Official Review · Reviewer_hRkg · 2024-05-21

**Rating:** 4
**Confidence:** 3

**Summary:**

This paper proposes a novel image captioning framework that integrates causal inference, addressing limitations in existing models. By incorporating causal concepts into training, the models can handle obscured or edited image content more easily. Through two variants, the approach improves generalization across datasets, reduces hallucinations, and enhances faithfulness to images. This innovation signifies a significant advancement in the interpretability and robustness of image captioning systems.

**Strengths:**

1) By providing counterfactually explainable descriptions, the framework advances the interpretability of image captioning tasks, fostering trust and understanding in AI-generated outputs.
2) By incorporating causal concepts, the proposed framework enhances the models' ability to handle scenarios involving obscured or edited image content, improving their robustness in real-world applications.
3) Sufficient in validation, the two variants presented in the framework enable better generalization across different datasets, making the approach applicable to a wide range of image captioning tasks and scenarios.

**Limitations:**

1) This paper seems quite similar to ‘Show, Deconfound and Tell: Image Captioning with Causal Inference’, but there is no reference to this paper. Can the authors state the difference between this paper and the proposed method?
2) I am curious that in table 1, it seems like NDE is working better than TE, but in factual images, the TE and NDE results seem not so as generalized as the counterfactual images. Is there any reason or explanations on this part?
3) For evaluation in biased datasets, are there any results on unbiased datasets, and see how the existing and current proposed methods differ in results?
4) In Figure 2, so the $Y_I$/effect is an unobserved counterfactual? The figure is not so clear to me.

**Suitability:**

3

---

### Official Review · Reviewer_vDvo · 2024-05-24

**Rating:** 4
**Confidence:** 4

**Summary:**

This paper proposes an image captioning framework based on counterfactual scenarios. The authors introduce causal graph into the model training, making the model more robust and interpretable, thereby mitigating the hallucination problem to some extent.

**Strengths:**

1. Authors propose a generic counterfactural regularize image captioning framework, which mark generated descriptions more human-like, explainable and robust.

2. Authors introduce causal graph into the training of image captioning, enhancing the better alignment between images and text.

3. The writing and examples in the paper make it easy to read.

**Limitations:**

1. In the process of generating counterfactual captions (Section 4.1), the initial model was not clearly specified. And I want to know how to ensure that hallucinations do not appear in counterfactual captions.

2. compared to other methods, the performance of proposed method is not significant in Flickr30k Entities and MSCOCO. But this is not the most important thing. In Table 3, the authors' method modeled the counterfactual data, but it only achieved about a 4% improvement in performance on the counterfactual images.

3. What are the advantages and disadvantages of TE and NED, respectively?

**Suitability:**

3

---

### Meta-Review · Area_Chair_7pt3 · 2024-07-04

**Recommendation:** Accept (Poster)
**Confidence:** 4

**Metareview:**

This work presents a framework using causal inference to enhance image captioning models for interventional tasks and counterfactual explanations, reducing hallucinations and increasing faithfulness to images across various datasets. Existing models often struggle with obscured or edited images, but the proposed approach addresses these challenges.

Three reviewers vote for acceptance, while Reviewer GZrh still has concerns. All reviewers acknowledge that the image captioning framework integrating causal inference can address limitations in existing models. Reviewer GZrh maintains concerns about the scalability of the proposed method, the evaluation metric, and its generalization ability on unseen and novel test sets. I agree with these concerns but, considering the contribution of using causal inference and the good empirical evaluation, I recommend accepting this submission. I suggest the authors provide a detailed discussion on these potential limitations or future directions. Please ensure the rebuttal (especially the newly included analysis) is well incorporated into the revised version.